# Current Insights on the Impact of Proteomics in Respiratory Allergies

**DOI:** 10.3390/ijms23105703

**Published:** 2022-05-20

**Authors:** Juan Carlos Vizuet-de-Rueda, Josaphat Miguel Montero-Vargas, Miguel Ángel Galván-Morales, Raúl Porras-Gutiérrez-de-Velasco, Luis M. Teran

**Affiliations:** Department of Immunogenetics and Allergy, Instituto Nacional de Enfermedades Respiratorias Ismael Cosío Villegas, Calzada de Tlalpan 4502, Sección XVI, Tlalpan, Ciudad de México 14080, Mexico; jos.miguel.montero@gmail.com (J.M.M.-V.); yumay0102@gmail.com (M.Á.G.-M.); raul.porras@cannapeutas.mx (R.P.-G.-d.-V.)

**Keywords:** proteomics, biomarkers, mass spectrometry, allergy, asthma, allergic rhinitis, aspirin-exacerbated respiratory disease, Pollen Food Allergic Syndrome, nasal polyps, airway inflammation

## Abstract

Respiratory allergies affect humans worldwide, causing extensive morbidity and mortality. They include allergic rhinitis (AR), asthma, pollen food allergy syndrome (PFAS), aspirin-exacerbated respiratory disease (AERD), and nasal polyps (NPs). The study of respiratory allergic diseases requires new technologies for early and accurate diagnosis and treatment. Omics technologies provide the tools required to investigate DNA, RNA, proteins, and other molecular determinants. These technologies include genomics, transcriptomics, proteomics, and metabolomics. However, proteomics is one of the main approaches to studying allergic disorders’ pathophysiology. Proteins are used to indicate normal biological processes, pathogenic processes, or pharmacologic responses to a therapeutic intervention. In this field, the principal goal of proteomics has been to discover new proteins and use them in precision medicine. Multiple technologies have been applied to proteomics, but that most used for identifying, quantifying, and profiling proteins is mass spectrometry (MS). Over the last few years, proteomics has enabled the establishment of several proteins for diagnosing and treating respiratory allergic diseases.

## 1. Introduction

The World Health Organization (WHO) estimates that around 25% of the world’s population suffers from respiratory allergic diseases [1]. Airborne allergens cause inflammation of the airways, and the most common allergens are house dust mites, pollen, proteins in animal hair, and animal urine. Air pollutants can aggravate allergy symptoms. The most important are particulate matter (PM_10_ and PM_2.5_), ozone (O_3_), nitrogen dioxide (NO_2_), carbon monoxide (CO), and sulfur dioxide (SO_2_), among others [1]. Pollutants penetrate the airways, triggering airway inflammation and exacerbating respiratory symptoms. Barrier dysfunction in the lung allows allergens and environmental pollutants to activate the epithelium further and produce cytokines that promote the induction and development of immune responses [2]. Therefore, respiratory allergies are more frequent in cities with high air pollution. Additionally, climate change extends the flowering period and pollen production of many tree species, resulting in chronic healthy affectations [3]. Pollen can also cause cross-allergies with some foods because they have similar proteins. For example, in oral allergy syndrome (OAS), people with a respiratory allergy who eat fresh and raw fruits and vegetables can suffer an allergic reaction in the lips, mouth, and throat [4]. Respiratory allergy is a type I hypersensitivity reaction mediated by IgE. The IgE-mediated mechanism involves a sensitization step in which Th2 cells produce cytokines such as IL-4, IL-5, and IL-13, which produce eosinophilia and induce specific IgE production. The IgE molecules bind to FcεRI receptors on mast cells (MCs) and basophils. This bind triggers a complex cascade signaling that leads to the release of inflammatory and vasoactive mediators such as histamine, leukotrienes, and vasopressin, among others, which cause the clinical response [5].

Biomarkers are defined as characteristics that are objectively measured and evaluated as indicators of normal biological processes, pathogenic processes, or pharmacologic responses to a therapeutic intervention. Clinical biomarkers offer some advantages: they are less expensive and usually measured quickly [6]. Unlike genes or transcripts, proteins are the most informative biomarker, are differentially expressed during disease states, and can undergo changes in protein folding and post-translational modifications relevant to understanding disease pathophysiology. Proteins can be measured and evaluated to compare the normal versus pathogenic biological processes or pharmacologic responses to develop therapeutic interventions. Mass spectrometry (MS) is the core technology used for current proteomics studies. It is helpful to discover new proteins as indicators of pathogenic processes or pharmacologic responses to treatment in allergenic diseases. In this review, we summarize the current insights of proteomics for the diagnosis and treatment of respiratory allergies.

## 2. MS-Based Proteomics

Proteomics studies the complete set of proteins present in a live organism at a specific time or condition, including expression, structure, functions, interactions, and modifications, which are crucial for early disease diagnosis, prognosis, and monitoring of disease development [7,8]. Although other techniques are relevant, MS has been the leading technology for proteomic analysis. As a result, the human proteome map was constructed employing MS [9,10]. With recent advances in instrumental devices, bioinformatics pipelines, and machine learning algorithms, proteomics has expanded to identify and analyze thousands of proteins with quantification capabilities [11]. MS determines the mass-to-charge (*m*/*z*) ratio of gas-phase ions produced in an ionization source such as in electrospray ionization (ESI) [12] and the matrix-assisted laser desorption ionization (MALDI) [13]. In addition, liquid chromatography (LC) coupled to tandem MS is the most common method to large-scale characterize proteins in complex biological samples [14,15,16].

In MS-based methods for proteomics, we can identify two approaches: bottom-up and top-down. Bottom-up is also called shotgun proteomics, and it is employed to identify proteins, post-translational modifications, and quantify biomarker discovery and diagnostic screening [17]. A mixture of proteins is enzymatically digested with a protease into mixtures of peptides before separating by LC. Then, the peptides are ionized and separated according to *m*/*z* in a first MS to be immediately split into fragmentation ions for the MS2 or MSn depending on the instrument capabilities. The mass spectra generated are compared with theoretical MS/MS patterns from databases and scores based on peptide-spectrum matches (PSMs). Additionally, de novo sequencing is possible. In this step, the typical analysis software includes MASCOT, SEQUEST, and X! Tandem [18]. Despite this approach being the most common method for proteome screening, there are limitations, including the fact that most proteins are identified based on few peptides, protein isoforms and post-translational modifications are often missed, and low-abundance proteins often will be lost or suppressed by other high-abundance proteins.

The other top-down MS approach can directly sequence proteins by LC-MS/MS. Here the intact proteins are chromatographically separated and detected directly without enzymatic digestion by ESI or MALDI. Then, ions generated in the ionization source are fragmented and analyzed in tandem mass spectrometry. This strategy provides more information on identifying and quantifying the protein isoforms, sequence variants, and post-translation modifications. However, routine identification of only the highest abundance proteins makes it difficult to characterize lower abundance proteins [19,20,21]. On the other hand, the quantification of proteins is crucial for understanding the complex biochemical mechanisms involved in a human disease condition. Protein levels in response to the environment, differential expression analysis, and protein–protein interaction reflect the body’s steady state. The proteomics techniques can be relative or absolute quantitation. The commonly used methods include label-free quantification (LFQ), the most widely used strategy for proteome quantification due to its simplicity and minimal interference. However, only relative quantification of proteins is possible with this methodology; no other biomolecule is added to the sample. It is usually employed in the clinical practice of searching biomarkers in cancer research when tumor versus normal tissues are compared [22]. Relative and absolute quantification is possible with stable isotope labeling with amino acids in cell culture (SILAC). Isotopes of Lys and Arg (13C or 15N) are added to the cell medium, labeling proteins for detection in MS1 by mass spectrometry. SILAC was used to find individual biomarkers in clinical practice [23].

Another quantitative method is isobaric tags for relative and absolute quantification (iTRAQ). It consists of comparing a reporter group of peptides with a balanced group. Qualitative and quantitative analysis can be performed simultaneously [24]. Another technique consists of the use of tandem mass tags (TMT). Labeling proteins with a reporter, normalizer, and an amine-reactive group allows analyzing different samples in a multiplex run with high precision and fewer missing values than LFQ [25]. Targeted proteomics detect low-abundance and specific proteins on multiple-, selected- or parallel-reaction monitoring (MRM, SRM, and PRM, respectively). These acquisition methods target specific peptide sequences and quantify protein isoforms and post-translational modifications, producing more reproducible and precise results [26]. An example of this quantitative technique is Absolute Quantification (AQUA), which incorporates synthetic peptides containing stable isotopes as internal standards. Then, the ratio between endogenous and the synthetic peptide is used to calculate the absolute quantitation of desired protein [27].

Besides MS, gel-based proteomics and immunological methods remain useful for allergen identification and respiratory illnesses. Two-dimensional electrophoresis (2-DE) consists of focusing proteins according to their isoelectric point (IEF) and by a molecular weight [28,29]. The presence or absence of spots provides valuable information about the dysregulation, level expression, quantity, and misshaping of proteins related to a respiratory disorder. However, 2-DE is technically laborious and challenging to replicate, and frequently more than one protein is in the same spot; thus, quantification is not precise. Additionally, western blotting is performed to detect IgE-reactive spots employing sera from allergic patients subsequently characterized by MS. Proteomics for respiratory allergies often depend on the sample type to be analyzed. These include blood cells, plasma, serum, sputum, bronchoalveolar and nasal lavage fluid (NLF), exhaled breath condensate, and biopsies of the lung and nasal polyps (NPs) [26]. Figure 1 shows the general overview of proteomics techniques for respiratory allergy. The use of proteomics for the study of respiratory allergies has seen a significant increase, focusing on the identification and structural characterization of allergens, protein profiling of the disease for biomarker discovery, in vitro diagnosis of allergen sensitization, and evaluation of the efficacy of immunotherapy [30,31,32].

## 3. Proteomics Approaches in Respiratory Allergy

Proteomics studies can increase sensitivity and specificity in allergic diagnosis and treatment. Novel diagnosis methods involve non- or minimally invasive local sampling, enabling an earlier patient diagnosis to distinguish infections from allergies. Altogether, all these findings illustrate the potential use of proteomics to identify new protein changes associated with allergies. The respiratory system is often divided into the upper (nose, mouth, larynx, pharynx, and trachea) and lower respiratory (bronchi, bronchioles, and lungs). Upper respiratory includes allergic respiratory diseases such as allergic rhinitis (AR), nasal polyps (NPs), and pollen food allergy syndrome (PFAS). Meanwhile, the lower respiratory includes asthma and aspirin-exacerbated respiratory disease (AERD), where bronchi and lungs are involved.

### 3.1. Upper Respiratory

#### 3.1.1. Allergic Rhinitis (AR)

AR has a variable prevalence in the general population. The clinical manifestations can help to distinguish between AR and rhinitis of mechanical origin. AR of the nasal mucosa presents with mucosal hyperemia, watery rhinorrhea, repeated sneezing, nasal obstruction, congestion, and occasional pollinosis. According to ARIA guidelines, AR is classified as seasonal (SAR) when it is caused by pollen antigens and occurs more intensely in spring and summer due to increased exposure to aeroallergens [33,34]. In AR, the common biomolecules studied are mRNA, metabolites, and proteins [35]. Research on proteomics in AR has shown that the principal proteins come from immune response or airway inflammation. Proteins provide complementary information on the status and severity of AR, which has allowed specific therapy to be given when these proteins disappear.

AR is treated generally with corticosteroids and antihistamines, the effect of the administration of which can be studied by employing NLF and proteomics. Wang (2011) provided a useful comparison of proteins in patients with seasonal allergic rhinitis (SAR) with low (LR) and high (HR) responses to corticosteroids treatment [36]. Employing LC-MS/MS, they found nineteen proteins for discrimination between HR and LR. The specific proteins found were Orosomucoid 1 and 2 (ORM), apolipoprotein H (APOH), fibrinogen alpha chain (FGA), cathepsin D (CTSD), and serpin peptidase inhibitor (SERPINB3). These proteins indicate that the treatment exerts systemic and local immunomodulation on the acute phase immune response in HR and not in LR. Both APOH and SERPINB3 are upregulated in AR, suggesting that APOH may be involved in an upregulated state in rhinitis, while SERPINB3 acts at different stages of pathogenesis [36]. SERPINB3 is upregulated in asthma patients and mediates mucus production [37], also in chronic obstructive pulmonary disease (COPD) [38]. SERPINB3 is physiologically involved in the regulation of differentiation in the normal squamous epithelium and is overexpressed in neoplastic tissue of epithelial origin, where it might be involved in the apoptotic pathway as a protease inhibitor. Serpins exhibit a unique mechanism of inhibition during which they undergo a dramatic conformational change to a more stable form [39]. It has been proposed that downregulation of SERPINB3 and SERPINB4 can control memory TH2 cells and helps to control grass pollen allergy [40].

The analysis of proteins in nasal secretions provides valuable information for studying nasal mucosa diseases. As mentioned before, NLF is an excellent biological source in AR, but novel sampling methods need to be developed. Lü and Esch described a method for collecting nasal secretions and analyzing their proteomic profile. The levels of immunoglobulins and the grass/weed pollen allergen-specific antibodies were 6- to 290-fold increased when the novel nasal secretion collector was utilized. Additionally, the concentrations of cytokines, eosinophil cationic protein (ECP), and tryptase in nasal secretions obtained by the nasal secretion collector were at least 8-fold higher than those tested in nasal lavages [41]. These results demonstrate that the sampling method is critical in proteomic experiments. In addition, the level of ECP is consistent with it being an important protein in eosinophil-mediated allergic inflammation such as asthma [42,43] and chronic rhinosinusitis [44].

The sinonasal epithelial barrier is comprised of tight and adherens junction proteins. Disruption of epithelial barrier function has been hypothesized to contribute to allergic diseases such as AR through the increased passage of antigens and exposure of underlying tissue to these stimuli [45,46]. Prior studies have not well-described proteins of transepithelial transport of allergens and AR-specific proteases and protease inhibitors. Tomazic’s informative study provides a valuable insight into proteins in nasal mucus in allergic rhinitis patients using LC-MS/MS. This group found principally five proteins markedly elevated in AR: apolipoprotein A-1 (APOA1), apolipoprotein A-2 (APOA2), and a2-macroglobulin (A2M), A1-antitrypsin (SERPINA1), and complement protein C3 (C3) [47]. Additionally, Tomazic et al., in 2015, also investigated the number of apolipoproteins in nasal mucus to validate their previous findings. The proteome of the nasal mucus of 22 patients (12 healthy and 10 with AR) was examined. This study confirms the presence of B100 protein, APOA1, and APOA2 in the mucosa of AR patients but not in healthy individuals. Finally, according to the author, these proteins present in mucus indicate that the immune response is local [48]. Subsequently, this same group identified changes in seasonal proteins of nasal mucus in AR. Eight proteins were abundant and differentially expressed in SAR, such as Clustering (CLU) and the Ig kappa C-chain region (IGKC). Meanwhile, glutathione S-transferase P 1 (GSTP1), neutrophil elastase (ELANE), histone H2B type 1-K (HIST1H2BK), and several S100 family proteins, including S100-A8 protein (S100A8) and S100-A12 protein (S100A12) were downregulated in SAR [49]. iTRAQ-based proteomic analysis reveals that SERPINA1 is also implicated in asthma [50] and allergic asthmatic populations sensitized to house dust mites [51]. Deficient SERPINA1 phenotypes have been related to the severity of asthma in adults [52]. APOA1 is a lipoprotein involved in lipid metabolism and has been recently related to chronic thromboembolic pulmonary hypertension [53], allergy, and skin diseases [54], and it is also proposed as a biomarker of anaphylaxis [55]. These two proteins can be an excellent candidate for the diagnosis of AR disease.

In addition, antimicrobial peptides (AMP) play an essential role in the immune response. In 2018, one comprehensive study was carried out by Preianò, who employed mesoporous silica particles (MPS) in combination with MALDI-TOF/TOF MS, which enriches nasal fluid in its antimicrobial peptides component, to discover new potential diagnostic markers of respiratory disorders. They found defensins, Statherin, β4-Thymosin, P-D peptide, β-MSP, SLPI, and lysozyme C. The authors proposed implementing mesoporous materials as a potential clinical and diagnostic utility [56]. Signal Transducer and Activator of Transcription (STAT) are transcription factors that mediate cellular immunity, proliferation, apoptosis, and differentiation in AR. Chen et al. investigated whether STAT3 regulates A2M expression in persistent AR. Employing iTRAQ and LC-MS/MS, about 133 differentially expressed proteins were identified, and the blood coagulation pathway is one well-associated pathway. The proteins differentially related to coagulation were principally A2M, F2 (prothrombin), plasminogen (PLG), coagulation factor XII (FXII), and coagulation factor III (TF or F3); meanwhile, the transcription factors STAT3 and A2M were detected in the nasal mucosa. These results suggest that A2M may be a key pathway in the regulatory network to the allergic inflammatory response and part of the pathogenesis in AR patients [57].

In the general adult population, about 2% have concomitant asthma, rhinitis, and eczema. Of sensitized adults, about 6% have coexistence of the three conditions [58]. To explain the coexistence of asthma, eczema, and rhinitis, a computational in silico analysis was performed by Aguilar et al. in 2017 on allergic diseases. They searched for proteins and associated genes of the functional interaction network (FIN) topology with diseases in Online Mendelian Inheritance in Man (OMIM) and UniProt. There were 15 pathways involved in the multimorbidity of asthma, eczema, and rhinitis, including IL4 signaling related to GATA3, which was the only one found in all three cases. The results suggest that IL-4-related GATA3 activates Th2 cytokine gene expression and is the most relevant [59]. GATA3 is a transcription factor that can stimulate IL-4, IL-5, and IL-3 secretion in T CD4+. This transcription factor can monitor Th2 response in AR [60].

Another significant aspect is that the pollen from grasses and trees can trigger AR mostly during the pollen season. Ndika (2017) offered an extensive analysis of the molecular elements related to AR during and after the pollen season using orbitrap-based quantitative proteomics, proteome identification, and quantification from LC-MS/MS analysis of differential abundance (DA) proteins and Western blot validation. Over 133 DA proteins were significantly overrepresented, including interferon 1 signaling. Cystatin (CST1) and myeloblastin, which protect against protease activity of allergens and have a role in epithelial barrier function, were differentially abundant in patients with SAR. Furthermore, compared with patients with ARS, healthy controls have a proteomic response to the stations, which may serve as a therapy for preventing AR [61]. CST1 was found as overexpressed in dog-sensitized children with severe allergic airway disease and can be used as a possible biomarker of the severity of allergic airway disease and a possible therapeutic target for the future treatment of airborne allergy [62].

Biomarkers for immunotherapy in patients responding and not responding to allergen immunotherapy treatment (AIT) have been assessed by LC-MS/MS. Ma and colleagues identified leukotriene A4 hydrolase (LTA4H) from the serum of patients who responded to immunotherapy. Serum LTA4H can predict the outcome of AIT [63]. A more straightforward and accurate test recently found differential biomarkers in serum from AR and asthma. Using a shotgun MS approach based on combinatorial ligand-peptide libraries and iTRAQ-LC-MS/MS, this group found that insulin-like growth factor binding protein (IGFBPs), and other proteins such as IGFALS, HSPG2, FCN2, or MASP1, are differentially expressed. They suggest that IGFALS can be used as a biomarker [24]. In summary, the important proteins associated with AR are mucus production, apoptotic pathway, coagulation, allergic inflammatory response, protection of epithelial protection, leukotriene synthesis, and transcription factors involved in IL production. The main proteins described in AR by proteomics are shown in Table 1.

#### 3.1.2. Nasal Polyps (NPs)

NPs are inflammatory masses that cause chronic nasal obstruction, usually associated with chronic rhinosinusitis (CRSwNP). NPs are transparent, pale-gray edematous projections originating from nasal ethmoidal mucosa near the middle turbinate. These are frequently associated with respiratory diseases, including AR and asthma, cystic fibrosis, AERD, and chronic rhinosinusitis (CRS). NPs are relatively common in the adult population, and they are rare in children under ten years old [64]. Exosomes are epithelial-derived vesicles containing conserved proteins representative of their parent cell and are secreted by all cell types into almost all body fluids, including nasal mucus [65]. Exosomes can be purified from the nasal mucus in a noninvasive way that will facilitate diagnosis and treatment. Several differentially expressed proteins involved in epithelial remodeling were found in exosomes secreted by human nasal epithelial cells [66]. These proteins influence cell proliferation, which is crucial to the remodeling of the sinonasal mucosa. Recently, in 2021, Wang et al. studied exosomes from the NLF of patients with NPs by MS. In this study, mucin 5AC (MUC5AC) was significantly up-regulated. The author proposed that this protein could be a potential therapeutic target of NPs [67]. MUC5AC is the major secreted polymeric mucins in airways, and their compositions affect mucus properties. Levels of MUC5AC and MUC5B were higher in patients with mild asthma [68]. Human neutrophil elastase induces MUC5AC in CRS through miR-146a [69]. These experiments show the relevance of mucins in allergic diseases. Several exosomal proteins of CRSwNP have been found as possible biomarkers like CST1, PRDX5, and GP6 [70]. Among them, CST1 and PRDX5 are more relevant as potential biomarkers in allergy. CST1 is a cysteine protease inhibitor that has been up-regulated in dog-sensitized children with severe allergic airways [62], and a bioinformatic analysis revealed a potential link between AR and asthma [71]. CST1 is also related in cancer [72,73]. Peroxiredoxins (PRDXs) are a family of six antioxidant proteins that may promote or inhibit carcinogenesis [74]. For example, PRDX5 has antitumor activity [75] and it was found as increased in an asthmatic murine model [76]. These findings suggest that PRDXs are potential biomarkers in allergic and cancer diseases. Furthermore, the coagulation pathway has been previously implicated in the etiopathogenesis of CRSwNP. Mueller et al. compared the proteomic profile between tissue and mucus-derived exosomes of CRSwNP where the coagulation pathway was altered. The most overexpressed proteins were fibronectin and fibrinogen gamma chains; however, the von Willebrand factor was downregulated. The von Willebrand factor was also decreased in late-phase response to allergen challenges in asthmatic people [77]. The von Willebrand factor helps platelets in the blood to clump together and stick to the wall of blood vessels, which is necessary for normal blood coagulation.

The interrelationship between human airway epithelium and complement proteins may affect airway defense, airway function, and airway epithelial integrity. Proteins involved in complement cascade pathways were found as enriched in the nasal mucus of individuals with CRSwNP compared with control subjects. Interestingly, Complement 3 (C3) was associated with disease severity [78]. C3 is an immune system protein that contributes to innate immunity and is released in response to proinflammatory cytokine stimulation in the bronchial epithelium, which might be a local airway defense regulatory mechanism. Eosinophils are multifunctional granulocytes capable of releasing various cytokines and chemokines in NPs of patients with eosinophilic rhinosinusitis (NP-EOS). A study conducted by Miyata et al. (2019) isolated eosinophils from patients with CRS for proteome analysis by LC-MS/MS. The results revealed that GGT5 was up-regulated, and DPEP2 was downregulated. GGT5 (gamma-glutamyltransferase 5) is an enzyme able to convert leukotriene C4 to leukotriene D4; meanwhile, DPEP2 (dipeptidase 2) hydrolyze a variety of dipeptides, including leukotriene D4. Changes in these enzymes appeared to define the inflammatory phenotype of NP-EOS with enhanced LTD4 production [79], making them potential therapeutic targets to control eosinophilic inflammatory diseases.

Another excellent noninvasive biological source is nasal secretions. Kim (2019) analyzed nasal secretions from patients with CRS and NPs by LC-MS/MS. Ten proteins were up-regulated in CRSwNP. The most relevant are FTL, FTH1, and GAA; meanwhile, fourteen proteins were downregulated, and the most important are: S100A7, SERPINB13, SERPINB8, CALML5, and RNASE3. Globally the IL signaling and neutrophil-mediated immune responses are increased in CRSwNP, and iron ion metabolism can be associated with CRS and NPs development [80]. SERPINs are serine protease inhibitors that play a crucial role in the fibrinolytic system related to chronic inflammatory lung diseases [81]. Meanwhile, S100A7 (S100 calcium-binding protein A7) is a member of the S100 family that has been involved in the regulation of blood-immune physiology and calcium homeostasis among many other functions, and is related to different subtypes of CRS [82].

Oral steroids have been traditionally used in the treatment of CRSwNP. However, the proteomic effect of its use has not been studied intensely. In 2020, Workman et al. analyzed the proteome of CRSwNP after steroids treatment using SOMAscan. Only 16 proteins were downregulated and 22 were up-regulated after treatment, highlighting that lactoperoxidase (LPO) and platelet factor 4 (PF4) were increased substantially [83]. LPO is a peroxidase enzyme that protects against bacteria and viruses; meanwhile, PF4 is a protein related to blood coagulation. The transcriptomic and proteomic analysis revealed that Serum amyloid A (SAA) is associated with a reduced response to oral corticosteroids. SAA levels may have potential value in predicting corticosteroid insensitivity in CRSwNP patients [84]. The evidence reviewed here shows that in NPs, the principal proteomics biomarkers are related to blood coagulation, lipid metabolism, and leukotrienes synthesis. In addition, exosomes and nasal secretions are excellent biological resources that can be analyzed in a noninvasive form. On the other hand, eosinophils play an important role in the inflammatory response, with modulation of immediate hypersensitivity reactions, defense against bacteria and parasites, and in particular the lipoxygenase pathway. Proteins found in NPs are shown in Table 2.

#### 3.1.3. Pollen Food Allergy Syndrome (PFAS)

PFAS is caused by the antigenic similarity between the pollen and food allergens such as uncooked fruit, nuts, raw vegetables, and flavors. Symptoms of PFAS may involve some organs like oral mucosa, skin, gastrointestinal, respiratory tracts, and the cardiovascular system. This type of allergy was initially called oral allergy syndrome (OAS), which affects the ear, nose, and throat. Some authors use both terms synonymously [85,86,87]. In PFAS, patients respond to various proteins. The prevalence of OAS for AR and asthma is ~9%, and in patients sensitized to pollen, the incidence ranged from 9.6% to 12.2% [88]. One to five percent of people suffer from food allergies, and clinicians report increasing numbers of PFAS. Several proteins are involved in this disease, like PR10 proteins, profilins, lipid transfer proteins, thaumatin-like proteins, isoflavone reductases, and β-1,3 glucanases. Three supplementary allergen families in PFAS were described in a recent review, in which oleosins, polygalacturonases, and gibberellin-regulated proteins were described [87]. A phosphoglyceromutase highly homologous to Api g 5 was identified by MS in cross-reactivity between mugwort pollen and fennel (a plant-derived food) [89]. Several gibberellins have been described as allergens in food allergy [90], and cypress pollen-sensitized patients [91,92]. Gibberellins are plant hormones that regulate the development process, including fruits and seeds. As mentioned in previous reports, GRPs have been described as allergens in plant-derived food [90]. These results highlight the importance of GRPs in vegetables and fruits that can produce cross-reactions in people sensitized to pollen.

These gibberellins can cross-react with peach allergen Pru p 7 [93] and orange allergy [94]. Another experiment with IgE immunoblots and MS was carried out in allergic Japanese children to compare GRPs (Gibberellin-regulated proteins) sensibilization to Japanese cedar pollen (Cryj). Nearly 50% of Japanese allergic to Cry j 7 are also allergic to GRPs present in fruits [95]. Pulp extracts are a source of allergenic proteins in food. Recently, in 2021, protein extracts from cherimoya pulp and peel were separated on 2D gels and incubated with serum from an allergic patient; 50-kDa reactive spots were identified as a glycosyltransferase. The biological function of glycosyltransferases is to catalyze glycosidic bond formation [96]. Several glycosyltransferases, like Fut7, are involved in allergic skin inflammation [97]. In another interesting study, a total of 38 Que i 1 sensitized patients were tested using Western blotting to determine the sensitization rate. Peptides from Que i 1 were analyzed by MAL-DI-TOF/TOF and Orbitrap LC-MS/MS. Que i 1 is responsible for PFAS caused by fruits [98]. Pathogenesis-related class 10 (PR10) proteins are highly conserved plant proteins induced in response to abiotic and biotic stress factors. Que i 1 is a PR10 family, and the allergen family has also been described in kiwifruit [99] and cashew nut [100]. To conclude this section, the literature identifies several proteins in PFAS that can cross-react with allergens which are pathogen-related proteins, profilins, gibberellins, and glycosyltransferases. A summary of the allergenic reaction in PFAS and the principal proteins identified by proteomics is shown in Table 3 and Figure 2.

### 3.2. Lower Respiratory

#### 3.2.1. Asthma

Asthma is the most common chronic airway inflammation that affects more than 300 million people worldwide. However, asthma pathogenesis is still poorly understood; the airway can be influenced by genetic, epigenetic, and multiple environmental factors like allergens, respiratory infections, exercise, atmospheric pollutants, smoke, and drugs [101]. The symptoms include breathlessness, cough, wheezing, chest tightness, and chronic airway inflammation [102]. Exposure to specific aeroallergens leads to immunological changes culminating in airway inflammation. Moreover, environmental irritants can exacerbate the disease and trigger asthma symptoms. The typical treatment of asthma has been based on the local and systemic application of bronchodilators and steroids to reduce inflammation [103].

Several studies have studied biological resources like the serum, sputum, saliva, bronchoalveolar lavage fluid (BALF), and bronchial biopsy [101,104]. Little is known about proteomics in the serum of asthmatic patients. Recently, SAA1 levels were measured in one hundred twenty-two asthmatic patients. SAA1 was higher in asthmatic patients, and this enrichment was found principally in patients with neutrophilic asthma [105]. SAA1 has been seen to interact directly with allergenic mite, Der p 13, and Blo t 13, and activates the SAA1-binding receptor, which drives type 2 immunity [106]. Additionally, SAA1 has been proposed as a biomarker of neutrophilic airway inflammation in adult asthmatic patients [105]. In children with asthma, proteins expressed differentially were studied in serum. This study identified 46 proteins differentially expressed, 12 up-regulated, and 34 down-regulated. These proteins were associated with the immune response, the inflammatory response, extracellular matrix degradation, and the nervous system [107]. Recently, in 2022, Weitoft and colleagues analyzed plasma samples after an allergen challenge by MS. They found 150 differentially regulated proteins, where the most up-regulated proteins were three protease inhibitors: alpha-1-antitrypsin (AAT), alpha-1-antichymotrypsin (ACT), and serine protease inhibitor (SERPINs), and the coagulation factor von Willebrand factor [108]. iTRAQ-based proteomic analysis reveals that SERPINA1 is also implicated in dust mite-related asthma [50]. SERPINA1 has also been described in AR as a potential protein for diagnosis [47]. SERPINs and coagulation factors seem to play an important role in allergic diseases.

Several enriched or depleted proteins that distinguish asthmatic patients were identified using sputum. Employing shotgun proteomics, ten proteins were found significantly up-regulated in asthma, including SERPINA1; meanwhile, seven proteins were significantly downregulated in asthmatics like S100A9, S100A8, SMR3B, and SCGB1A1. These proteins are related to defense response, inflammation, and protease inhibitory activity. The protein profile in patients with controlled (CA) and uncontrolled asthma (UA) revealed an increase in human neutrophil peptide-2, S100A9, β-amylase, neutrophil gelatinase-associated lipocalin, 4-aminobutyrate transaminase, and cystatin SA in patients with UA. Meanwhile, S100 calcium-binding protein A9 (S100A9) was up-regulated in neutrophilic sputum from patients with severe UA. These findings differ from those found by Gharib [109]. S100A9 is proposed as a biomarker of neutrophilic asthma-related to airway inflammation and steroid resistance [110,111]. Some S100 proteins have been found in patients with AR and nasal inflammation [112]. Sputum from patients with various airway diseases (chronic bronchitis, COPD, and asthma) was evaluated. In asthmatic people, Lactotransferrin (LTF), an antimicrobial; MUC5AC, Mucin 5B (MUC5B), BPI fold-containing family B member 1(BPIFB1), and Protein 14.3.3 (SFN) were downregulated. Meanwhile, Calmodulin 3 (CALM3) was upregulated [113]. In sputum, various inflammatory mediators were identified. Cytokines, chemokines, and growth factors were increased in severe asthma, primarily with increased neutrophils [114]. Therefore, MUC5AC is proposed as a potential biomarker [115], but these results seem controversial and must be reevaluated [116]. Nevertheless, MUC5AC has been found as overexpressed in patients with CRS [69,117,118].

Saliva is another body fluid measured in a noninvasive way in asthma [119]. Children’s saliva was analyzed by immunoassays for cytokine quantification. Cytokines profiles were associated with the asthmatic phenotype [120]. Okazaki studied the levels of the salivary surfactant protein D (SP-D) in asthmatic children. They found that salivary SP-D levels are increased in asthmatic children. They commented that salivary SP-D may reflect asthmatic inflammation in peripheral small airways and may be a useful marker for monitoring the degree of exacerbation in childhood asthma [121]. This is in contrast to earlier findings, in which SP-D levels in BAL fluids were lower in adults with severe asthma [122]. A recent comparative proteomic analysis using shotgun proteomics in asthmatic people revealed novel proteomic biomarkers in saliva. This study found several proteins in the salivary sample related to uncontrolled asthma: polycystic kidney and hepatic disease 1 (PKHD1)/fibrocystin, zinc finger protein 263 (ZNF263), uncharacterized LOC101060047 (ENSG00000268865), desmoglein 2 (DSG2), and S100 calcium-binding protein A2 (S100A2) [123].

A more recent study on children with severe asthma found that BAL cytokine/chemokine expression protein profiles were associated with refractory asthma and neutrophilic inflammation. Cytokines were overexpressed in neutrophilia asthma compared with no-neutrophilia asthma [124]. Another crucial biological source in asthma is a bronchial biopsy. In a proteome analysis with nanoLC-LTQ Orbitrap mass spectrometer in a bronchial biopsy, ANXA5 (Annexin A5 protein), DPT, HIST1H2AH, LMNA, PPIA, RPBL7, and RPBL8 were identified. These proteins are involved in cellular movement and immune cell trafficking; their role includes collagen fibrillogenesis, protein elongation, and chemotaxis [125]. ANXA5 and ANXA1 (Annexin A1 protein) have also been proposed as potential biomarkers in asthma [126,127]. ANXA5 is a soluble protein that binds to biological membranes and promotes its reparation [128]. Some proteomic experiments have been carried out to distinguish between different respiratory diseases. In 2021, Winter et al. investigated possible biomarkers for chronic obstructive pulmonary disease (COPD) and asthma. Only four biomarkers were analyzed with immunoassay methods: A2M, ceruloplasmin, haptoglobin, and hemopexin. Only hemopexin was found to distinguish between COPD and asthmatic patients, and it is suggested as a potential biomarker [129].

In addition, recent proteomics studies have been focused on finding biomarkers during treatment. Landi, in 2021, employed serum from severe asthma patients after benralizumab and mepolizumab treatment. They demonstrated APOA1 oxidation after mepolizumab treatment and ceruloplasmin and transthyretin oxidation with benralizumab treatment [130]. Since oxidative stress plays an important role in severe asthma, these proteins may regulate oxidative stress and maintain homeostasis. For example, ceruloplasmin acts as a circulating scavenger of superoxide anion radicals protecting cells and tissues from the effects of free radicals. A similar experiment was performed in serum protein profiles of patients with severe eosinophilic asthma before and after anti-IL5 or anti-IL5R therapies. CAYP1, A1AT, and A2M were found to be differentially expressed and could be used as potential biomarkers for treatment [131]. It is well known that smoke can increase asthma severity. Proteomic analysis of sputum supernatants of a cohort of asthma patients containing current-smokers (CSA), ex-smokers (ESA), nonsmokers (NSA), and healthy controls were examined. In this study, SOMAscan revealed a difference in the sputum proteome between NSA and CSA (CSF2, AGR2, and CXCL8), and between NSA and ESA (AZU1, ELANE, CFP, and CXCL8) subjects, with CXCL8 not discriminating between ESA and CSA [132]. Asthma phenotypes were predicted through cytokine/chemokine expression patterns using bronchoalveolar lavage (BAL) [133,134]. On the contrary, little is known about the health effects of electronic cigarettes (E-Cigarettes) in individuals with respiratory diseases such as asthma. A relationship between asthma and E-Cigarettes has been documented [135,136,137,138]. Menthol has been proposed as a contributor to respiratory diseases. Proteomics data showed upregulation of the oxidative stress pathway [139]. In order to study the effect of E-Cigarettes exposure, Ghosh and colleagues determined the bronchial epithelial proteome in humans after chronic exposure. Approximately 300 proteins were found differentially expressed. CYP1B1, MUC5AC, and MUC4 were increased in vapers. These proteomic studies in smokers reveal the important role of mucins, as well as in allergic diseases. This section attempted to briefly summarize the literature relating to proteins involved in asthma in different pathways like an immune and inflammatory response, coagulation, protease inhibitors, mucus production, calcium binding, cellular movement, immune cell trafficking, membrane reparation, and cytokines production. Proteins found in the diagnosis and treatment of asthma are shown in Table 4.

#### 3.2.2. Aspirin Exacerbated Respiratory Disease (AERD)

Aspirin-exacerbated respiratory disease (AERD) is a combination of disorders characterized by worsening eosinophilic rhinosinusitis, chronic hyperplastic sinusitis, nasal polyposis, and bronchial asthma caused by hypersensitivity to non-steroidal anti-inflammatory drugs (NSAIDs). This disease affects the upper and lower respiratory tract with epithelial alterations, production of cytokines, chemokines, and other molecules of immunity and inflammation. Overproduction of cysteinyl-leukotrienes (CysLT) via LTC4 synthase is hypothesized to be a critical pathway in the pathogenesis of AERD [140]. Up to 64% of AERD patients are allergic to common aeroallergens. Evidence for the involvement of IgE in AERD derives from a clinical study showing that the anti-IgE drug omalizumab suppresses the reaction to aspirin challenge in 62.5% of AERD patients who developed tolerance after three months of treatment. Urinary LTE4 levels, peripheral eosinophil counts, and serum periostin levels were significantly suppressed within seven days of omalizumab treatment [141]. The assessment of LTE4 and other eicosanoids is valuable and informative if performed in induced sputum supernatant [142]. Urinary LTE4 levels have been used to predict AERD in patients with asthma [143]. Additionally, the LTE4 level is a potential biomarker for chronic rhinosinusitis [144].

In 2013, Choi and co-workers examined proteins involved in AERD inflammation. Three candidate proteins of eosinophil activation were proposed for diagnosis: APOA1, A2M, and ceruloplasmin. In nasal challenge with lysine aspirin, eosinophil cationic protein and CysLT levels increased significantly during the early response to AERD. Plasma apolipoprotein H levels are different between aspirin-induced respiratory diseases and aspirin-tolerant asthma (ATA); this protein is proposed as a biomarker in AERD [145]. In severe eosinophilic asthma, A2M was differentially expressed after mepolizumab and benralizumab treatment [124]. In AERD, after administration of NSAIDs in asthmatic patients, bronchoconstriction and nasal polyposis are developed. It is important to distinguish between AERD and ATA patients for correct treatment. Immunohistochemistry for lipid-protein binding, 2-DE, and LC-MS/MS allowed the identification of 15 up-regulated proteins in AERD compared to ATA patients. Fatty acid-binding protein 1 (FABP1) expression in immunoblotting and immunohistochemical analysis was significantly higher in nasal polyps from AERD patients. Additionally, FABP1 was present in the epithelium eosinophils, macrophages, and smooth muscle cells of polyp blood vessels. There is a correlation between FABP1 and the development of AERD [146]. Fatty acid-binding proteins (FABPs) are abundant intracellular proteins that bind long-chain fatty acids (FA) and have been related to immune metabolic diseases [147]. Miyake studied epithelial protease inhibitors such as cystatins in 2019. They identified seven groups among patients with high levels of cystatin-2 NMDE (nasal mucosa-derived exosomes) in patients with AERD. The authors suggested that high levels of cystatin-2 NMDE predict the phenotype and severity of AERD in a non-invasive biopsy with potential pathophysiological diagnosis and differentiation of pre-existing diseases. Cystatin-2 levels predict CRS phenotype and disease severity in chronic rhinosinusitis [148]. The evidence reviewed here shows that in AERD, the proteins are related principally to cysteinyl leukotrienes synthesis, protease inhibition, and fatty acid binding. The principal proteins found in the diagnosis of AERD are shown in Table 5.

## 4. Proteomics in Allergy Diagnostics and Therapeutics

Skin tests are the most common method for allergy diagnosis using native allergen extracts. However, most allergen extracts are a mixture of complex and well-defined mixtures of many non-allergenic and allergenic components, making it challenging to identify the disease-eliciting allergen. Recombinant allergens have been produced to ensure better standardization, which can be used for diagnostics and AIT. For example, these allergens can be used in molecular-based allergy diagnostics, which allows for defining the allergen sensitization of a patient at the molecular level through purified or recombinant allergen. While these diagnostic methods may be very useful for detecting known allergens, they may have some limitations when allergens causing sensitization are unknown or when new sensitization proteins are present. Proteomics plays an important role here because it can identify any protein. Indeed, using an immunoproteomics approach, we identified novel allergens derived from Ligustrum, Red Oak, and Pecan [101,102,103]. Similar findings have been reported by other groups who have reported allergens derived from coconut, amaranth, buckwheat, and celery [104,105,106]. Immunoproteomics involves different techniques such as gels array-based techniques, MS, and in silico studies of their interactions with the immune system; the assessment of the allergens starts with allergen detection by serum from sensitized patients, followed by the characterization of the recognized proteins using MS. The MS has also been used to characterize natural extracts at pharmaceutical grade for allergen-specific immunotherapy [107] and monitor biomarkers in response to AIT. Using quantitative proteomics of pre-treatment sera derived from patients suffering from grass pollen allergy showed that high levels of O-glycosylated sialylated Fetuin-A isoforms correlated with a reduction of rhinoconjunctivitis symptoms after sublingual immunotherapy [108]. Immune epitopes derived from allergens can also be investigated using X-ray diffraction or hydrogen-deuterium exchange mass spectrometry (HDX-MS).

## 5. Conclusions

The incidence and prevalence of allergic respiratory diseases are increasing worldwide. However, no cure for these diseases has been found. One of the main objectives of using proteomics in allergic disease has been to identify proteins that can provide insights into the pathogenesis of the disease. The proteins identified for treatment or diagnosis of allergic respiratory diseases discussed in this review are shown in Figure 3. Indeed, proteomics has made it possible to characterize thousands of proteins in one single analysis, and it has allowed the identification of novel allergens which can be used in serodiagnosis. Proteomics is also useful for characterizing natural extracts at pharmaceutical grade for allergen-specific immunotherapy, and it can also monitor biomarkers in response to monoclonal therapy and AIT. It is expected that this technology may facilitate the development of personalized immunotherapeutics for allergic patients in the future.

## Figures and Tables

**Figure 1 ijms-23-05703-f001:**
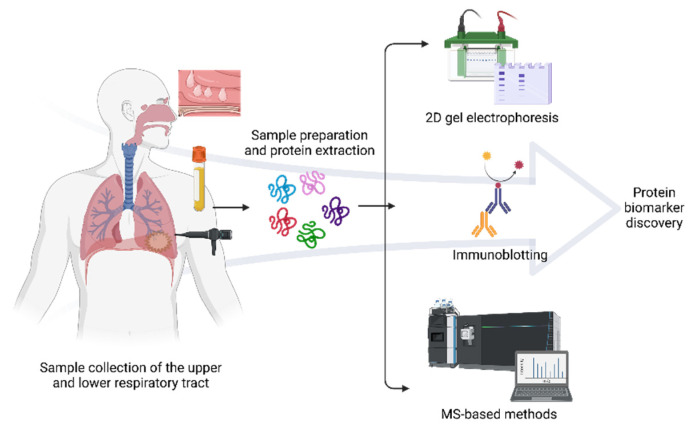
General proteomics workflow in respiratory allergy. Protein analysis of several biological samples such as serum, blood cells, NLF, BAL, NP, sputum, and saliva may reflect how the proteome varies in allergic diseases. The predominant methods for discovering protein biomarkers of allergy include 2D gel electrophoresis combined with immunochemical detection and subsequent identification by mass spectrometry. Created with BioRender.com (accessed on 10 May 2022).

**Figure 2 ijms-23-05703-f002:**
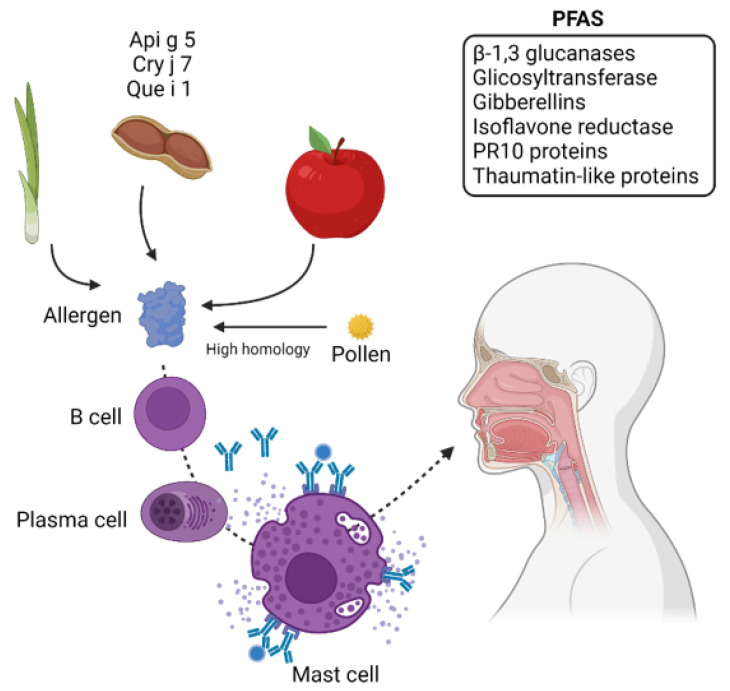
Potential proteins discovered in PFAS. PFAS causes allergies in people who are sensitized to pollen proteins. The high homology between allergy pollen proteins and proteins from food can cause cross-reaction, which is IgE-mediated. Allergens found in fruits and vegetables like Api g 5, Cry j 7, and Que i 1 and some proteins that can cause cross-reaction are also indicated. Gibberellins and PR10 proteins are common proteins that can cause cross-reactivity with some food allergens. Created with BioRender.com (accessed on 10 May 2022).

**Figure 3 ijms-23-05703-f003:**
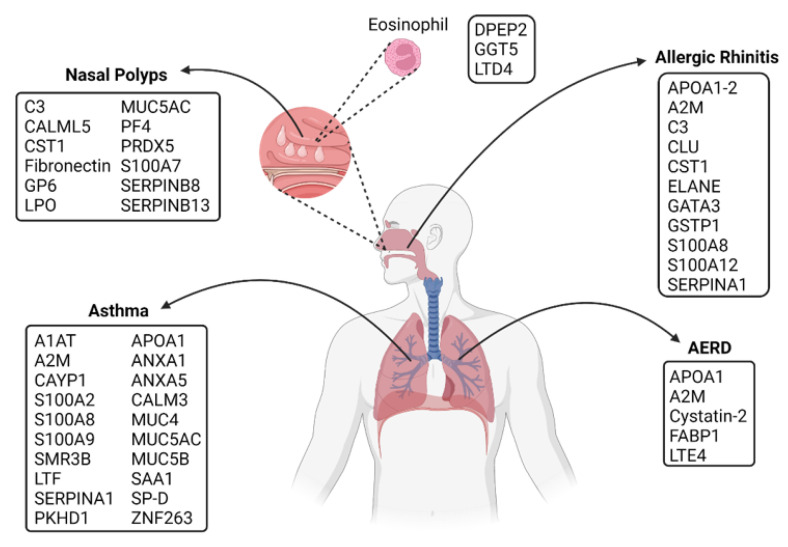
Potential diagnostic proteins found by proteomic technologies in upper and lower respiratory allergies. A non-exhaustive list of potential proteins described in diagnosis and treatment for asthma, nasal polyps, allergic rhinitis, and AERD. The same proteins are found in various allergy diseases like A2M, APOA,1 C3, CST1, SERPINs, some members of the S100 family, and Mucins. The principal pathways involved in these allergic diseases are mucus production, coagulation, inflammatory response, protease inhibition, cytokines synthesis, and membrane reparation. Created with BioRender.com (accessed on 10 May 2022).

**Table 1 ijms-23-05703-t001:** A non-exhaustive list of current proteomics technologies for Allergic Rhinitis diagnosis. Up arrows indicate up-regulated proteins; meanwhile, down arrows show down-regulated proteins.

Source	Biomarker	Proteomic Technology	Reference
NLF before treatment with glucocorticoids	↓ORM, APOH, FGA, CTSD, SERPINB3	LC-MS/MS	[41]
Nasal lavages	HSA, ECP, tryptase, cytokines, and total Igs	Bio-Plex suspension array Luminex xMAP system ELISA	[41]
Nasal mucus	↑ALB, IgA, BPIFBI, APOA2, A2M, APOA1, SERPINA1 and C3 ↓LTF, LYZ, SLPI, WFDC2, IGJ, Ig hc V-III region BRO	LC-MS/MS	[47]
Nasal mucus	↑APOA1, APOA2, APOA4 and B-100	LC-MS/MS	[41]
Nasal mucus (Pollen season)	↑CLU, IGKC ↓GSTP1, ELANE, HIST1H2BK, S100A8, S100A12 and ARHGDIB	LC-MS/MS	[41]
Nasal fluids (NF)	HNPs, Statherin, Thymosin-β4, P-D peptide, II-2, β-MSP, SLPI, Lysozyme-C, and their proteo-form	LC-MS/MS.	[56]
Serum	A2M, STAT3, p-STAT3 and IL-17	iTRAQ, SCX, and LC-MS/MS.	[57]
Nasal brush samples	CST1, PRNT3, IFIT1, IFIT3	Orbitrap-based, bottom-up label-free quantitative proteomic. LC-MS/MS	[41]
Serum after allergen immunotherapy (AIT)	LTA4H	Nanoflow (LC-MS/MS)	[41]

**Table 2 ijms-23-05703-t002:** Principal proteins found by proteomics involved in Nasal Polyps.

Source	Biomarker	Proteomic Technology	Reference
Exosomes from the NLF	MUC5AC and MUC5B	LC-MS/MS	[67,68]
	CST1, PRDX5, and GP6	SOMAscan^TM^	[70]
	Before steroid treatment ↓LPO, CAIII, PF4, PLAT ↑α2AP, LILRB2, CD209, APOE2	SOMAscan^TM^	[83]
	After steroid treatment ↑APOL1, CSNK2A2, LPO, ANG		
Eosinophils from nasal polyps	↑GGT5 ↓DPEP2	LC-MS/MS	[79]
Nasal secretions	↑FTL, FTH1, GAA ↓ S100A7, SERPINB13, SERPINB8, CALML5, and RNASE3	LC-MS/MS	[80]

**Table 3 ijms-23-05703-t003:** Principal allergens and cross-reactive proteins are described with proteomics in PFAS.

Allergen	Cross-Reaction Protein	Food	Proteomic Technology	Reference
Api g 5 (Celery)	Phosphoglyceromutase	Mugwort Fennel	Immunoblots and MS	[89]
Cry j 7 (Japanese cedar)	Gibberellins	Peach, citrus, and apple	2-DE and MS	[95]
Que i 1	PR10 family	Banana, melon, apple, watermelon, pear, kiwi	MALDI-TOF/TOF and Orbitrap LC-MSMS	[98]

**Table 4 ijms-23-05703-t004:** Proteins are involved in the diagnosis and treatment of asthma. Up arrows exhibit up-regulated proteins; meanwhile, down arrows show down-regulated proteins.

Source	Biomarker	Proteomic Technology	Reference
Serum	↑IGKV2-40, IGHV3-74, IGKV1-27, V1-19, IGLC-7, APP, IGKV1-16, PIP ↓APOD, ACAN, CNTN1, C1S, AOC3, LRP1, COL10A1, ITGB1, PTPRG, ADAMTS13, DPP4, IFNa2, HSPA1A, APOB, NCAM2, TNXB, ACTB, CACNA2D1, POSTN, ALP, PK, LTF, ELANE, CTSG, MPO, G6PD, PFN1epl	LC-MS/MS	[107]
Serum during treatment	Mepolizumab: APOA1, CAYP1, A1AT and A2M	SWISS2DPAGE software	[131]
	Benralizumab: CERU, CAYP1, A1AT, and A2M		
Sputum	↑HP, SERPINA1, PR4. ↓S100A9, S100A8, IGL, HTN1, SCGB1A1, SMR3B	LC-MS/MS	[109]
	↑CALM3 ↓LTF, MUC5AC, MUC5B, BPIFB, SFN	LC-MS/MS	[113]
	CSA-NH/NSA-NH: CSF, CXCL8, AGR ESA-NH/NSA-NH: AZU, ELANE, CFP, CXCL	SOMAscan^TM^	[132]
Saliva	↑SP-D	ELISA	[121]
	PKHD1, ZNF263, DSG2, S100A2	Shotgun proteomics	[123]
Endobronchial biopsies	ANXA5, DPT, HIST1H2AH, LMNA, PPIA, RPBL7, and RPBL8	NanoLC-LTQ Orbitrap mass spectrometer	[125]
Bronchial epithelial after chronic E-Cigarette exposure	CYP1B1, MUC5AC, and MUC4	LC-MS/MS	[116]

**Table 5 ijms-23-05703-t005:** A list of current proteins described in AERD by proteomic technologies.

Source	Biomarker	Proteomic Technology	Reference
NLF	ApoA1, A2M, and CP	MALDI-TOF/TOF	[145]
Nasal polyps samples	FABP1	Nano LC-MS/MS	[146]
Nasal mucosa-derived exosomes	Cystatin-2	ELISA-relative concentration	[148]

## Data Availability

Not applicable.

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
