# Peer review of "Current Insights on the Impact of Proteomics in Respiratory Allergies"

_ijms, 2022, doi:10.3390/ijms23105703_

Round 1
Reviewer 1 Report
- Please follow an upgrade in the last paragraph of the introduction given with proper citation.
“Biomarkers are defined as a characteristic that is objectively measured and evaluated as an indicator of normal biological processes, pathogenic processes, or pharmacologic responses to a therapeutic intervention. Clinical biomarkers offer some advantages: they are less expensive and usually measured quickly [https://doi.org/10.1016/j.cca.2020.04.015].”
Author Response
The last paragraph was updated according to the reviewer´s comment 1. Lines 55-58.

Reviewer 2 Report
- The review presents a summary of the subject but does not properly address the issue at hand. This text requires extensive revisions and reorganisation. The topic's title and the content presented inside it do not correspond at all.
- Example: The topic explicitly defines the insights of proteomics, yet the abstract describes the usage of all omics technologies; this entirely confuses the readership.
- Figure 1 is quite generalised and provides little information on the subject. However, this knowledge is very fundamental for proteomics and may be used to any biological inquiry.
- Table 1 is likewise quite brief and lacks adequate information. There are plenty further research available on the issue. Authors should present research fully with more details (In terms of including more columns).
- Similar considerations apply to Table 2.
- There is no information provided in image 2. It seems that the writers illustrated the figure with a biorender just for the purpose of giving a figure and paid little attention to the biological topic. In addition, the authors make no mention of the biorender programme anywhere in the whole text. Regarding the biorender policy, I believe it is essential that authors include a citation for the biorender.
- Page 13 is totally blank and the typesetting and layout have not been given much consideration.
- The S100 family is shown in Figure 3. It is peculiar that the authors want to assert that every isoform of this family is a candidate diagnostic protein. There are over twenty-four well-known isoforms.
Author Response
1-2. Reply. The title of the topic was changed according to the reviewer's comment. However, contrary to the reviewer's comment on the term omics, we consider that it should be kept because the idea focuses on the role of proteomics in the general context of the omics sciences.
3. Reply: Figure 1 and its caption were updated to provide more information about the general workflow on proteomics for allergy. Lines 146-150.
4-5. Reply: We consider that Tables 1 and 2 include the information necessary according to the review´s topic. More columns could make the tables difficult to read and understand.
6. Reply: Figure 2 provides the text and the figure´s caption information. Biorender is now cited in the figures, and we have permission for publication.
7. Reply: Page 13 has text. The typesetting was done according to the journal´s requirements.
8. Reply: In Figure 3, the exact names of S100 proteins were added to be more accurate.

Reviewer 3 Report
This version of the aritcle is much better, at least in my view.
Two minor comments/facultative suggestions only:
C1. Maybe I overlooked something but also saliva should be mentioned (PMID: 33925009).
C2. If you discuss LTE4 in AERD anyway, please, mention that the assessment of LTE4 and other eicosanoids is valuebale and informative also if performed in iduced sputum supernatant (PMID: 35330446).
Author Response
1. Reply: According to the reviewer’s comment #1 the most relevant results on proteomics analysis of saliva in asthmatics were added to the final version of the manuscript (Lines 428, 472-485.). In addition, Table 4 and Figure 3 were updated.
2. Reply: The article recommended by the reviewer is considered in the final version of the manuscript, and now we have mentioned it. Lines 553-554.

Round 2
Reviewer 2 Report
No further comments
This manuscript is a resubmission of an earlier submission. The following is a list of the peer review reports and author responses from that submission.
Round 1
Reviewer 1 Report
The subject is intriguing since it reflects the use of proteomics techniques in the study of respiratory allergies. Unfortunately, in this current review, there is insufficient research demonstrating the relevance of proteomics techniques, and hence the review falls short of giving key information relevant to the review article's subject. Authors should be capable of compiling an excellent review; nonetheless, this review should include thorough illustrations and tables. None of them is included in the review's current form. Above all, the review requires substantial reorganization, since many of the paragraphs transition seamlessly from one issue to the next.
Reviewer 2 Report
Respiratory allergies are currently severe problems still counting among the worldwide population. Therefore, it certainly does deserve attention to be studied.
Proteomics is an efficient omics tool that can bring important information about the molecular mechanisms based on proteome changes referred to DEPs, novel proteome biomarkers among others (PTMs, PPIs etc.)
This work deals with the advances in proteomic analyses of respiratory allergies such as AR, NPs, PFAS or AERD.
REMARKS
- Please follow an upgrade in the last paragraph of the introduction given with proper citation.
“Biomarkers are defined as a characteristic that is objectively measured and evaluated as an indicator of normal biological processes, pathogenic processes, or pharmacologic responses to a therapeutic intervention. Clinical biomarkers offer some advantages: they are less expensive and usually measured quickly [https://doi.org/10.1016/j.cca.2020.04.015].”
- Please, draw detail a general proteomic workflow used for proteome study of respiratory allergies, and provide it as a figure. It should contain a collection of samples, sample preparation and clean up, analysis and bioinformatic assessment.
- Please, provide a comprehensive comparison table where you mention all important data about the outcomes obtained from studies. The columns that the table should contain such as sample type, allergy type, number of patients enrolled, type of proteomic approach, biomarkers found, number of proteins identified etc.
- In conclusion, please provide information of authors´ future research aims in this field, if there are any
Reviewer 3 Report
I read the manuscript ijms-1655511.
Overall, very in general, the idea behind this manuscript is good and so is ist overall structure.
However, within the chapters there is no flow and no clear plan is followed. Different molecules are listed one after the other without any logical order, within short sentences not clarifying anything. Different things are mixed up with each other, such as e.g. exosomes pop up in the section on nasal polyps along with eicosanoids and some molecules oft he other type. Some summary is provided by the figures that are graphically nice but limited to the lists of molecules with no additional information and the corresponding parts of the text do not explain the things sufficiently.
What is clearly seen throughout the manuscript ist hat the knowledge presented in it is not systematic as not based on the systematic literature search or a a clear plan, e.g. presenting the most important things based on certain criteria. In other words, the contents of this work seem accidental, just a bit of this and that, including substance not being proteins such as eicosanoids.
This draft does not add much to our knowledge, does not make it systematic, does not sort it.